# Modulation of AMPA Receptors by Nitric Oxide in Nerve Cells

**DOI:** 10.3390/ijms21030981

**Published:** 2020-02-01

**Authors:** Violetta O. Ivanova, Pavel M. Balaban, Natalia V. Bal

**Affiliations:** Cellular Neurobiology of Learning Lab, Institute of Higher Nervous Activity and Neurophysiology of the Russian Academy of Science, Moscow 117485, Russia; pmbalaban@gmail.com (P.M.B.); bal_nv@mail.ru (N.V.B.)

**Keywords:** nitric oxide, AMPA, glutamate receptor

## Abstract

Nitric oxide (NO) is a gaseous molecule with a large number of functions in living tissue. In the brain, NO participates in numerous intracellular mechanisms, including synaptic plasticity and cell homeostasis. NO elicits synaptic changes both through various multi-chain cascades and through direct nitrosylation of targeted proteins. Along with the *N*-methyl-d-aspartate (NMDA) glutamate receptors, one of the key components in synaptic functioning are α-amino-3-hydroxy-5-methyl-4-isoxazole propionate (AMPA) receptors—the main target for long-term modifications of synaptic effectivity. AMPA receptors have been shown to participate in most of the functions important for neuronal activity, including memory formation. Interactions of NO and AMPA receptors were observed in important phenomena, such as glutamatergic excitotoxicity in retinal cells, synaptic plasticity, and neuropathologies. This review focuses on existing findings that concern pathways by which NO interacts with AMPA receptors, influences properties of different subunits of AMPA receptors, and regulates the receptors’ surface expression.

## 1. Introduction

Nitric oxide (NO) was established as a novel type of signaling molecule in the central nervous system (CNS) about 30 years ago [1], shortly after being recognized as the endothelium-derived relaxing factor in blood vessels [2,3]. In the CNS, NO participates in numerous functions, including learning and memory, sleep, feeding, movement, pain, anxiety, and reproductive activity [4,5,6]. NO plays multiple roles in the nervous system, including reciprocal functions (cytotoxic and cytoprotective effects), and is called a Janus molecule in the literature [7]. It is able to interact with many intracellular targets to trigger an array of signal transduction pathways, resulting in the activation or inhibition of output signals. At the same time, NO becomes toxic if it is produced in excess [8]. If a cell is in a pro-oxidant state, NO can undergo redox reactions to form toxic compounds, which cause cellular damage [8,9]. Since NO is involved in many pathways, it is likely to be involved in several diseases, such as hypertension, coronary heart disease, Alzheimer’s disease, stroke, and cancer [10].

Both its role as a transmitter molecule and the neurotoxicity of NO can be mediated through various receptors, including ionotropic glutamate receptors. All glutamate ionotropic receptors have a tetrameric structure and are permeable to cations. There are three major groups: α-amino-3-hydroxy-5-methyl-4-isoxazole propionate (AMPAR), *N*-methyl-d-aspartate (NMDAR), and kainate receptors. Relatively little information has been published regarding the interactions of NO with kainate receptors, in contrast to the AMPA and NMDA receptors with which NO interacts and determines many physiological functions in nerve cells, including synaptic plasticity. Synapses of brain cells can undergo a prolonged increase and decrease in connection strength in response to various types of stimulation and/or biochemical influences. It is believed that these processes, known as long-term potentiation (LTP) and long-term depression (LTD), underlie the formation of memory at the synaptic level. Both LTP and LTD can occur in the same cell, which allows dynamic control of the strength of synapses and may influence the storage of information in neural circuits.

NMDA channels are extremely important for many cellular processes, and at the same time, their excessive activation can lead to cell death. To avoid this, various molecules modulate NMDA receptors. One of the negative feedback mechanisms that prevents excessive activation of NMDA channels is modulation by NO. There are several comprehensive reviews describing NMDA modulation by NO [11,12,13]. In this report, we focus on AMPA receptors and the regulation of their subunits, mostly GluA1 and GluA2, by NO via various molecular pathways in neurons. We also discuss the NO–AMPARs interactions in the retina since this cooperation reflects the multifunctionality of NO well even in such a strictly specialized group of cells.

## 2. Nitric Oxide

NO is a gaseous neurotransmitter that, together with other signaling molecules, is involved in intercellular communication, and regulates intracellular processes. Due to its physicochemical characteristics, this small molecule can freely flow through membranes. NO is synthesized from L-arginine using the enzyme NO synthase; another product of this reaction is L-citrulline [14].

There are three isoforms of NO synthase—neuronal (nNOS or NOS1), endothelial (eNOS or NOS3), and inducible (iNOS or NOS2). eNOS is primarily responsible for the generation of NO in the vascular endothelium [15]. iNOS produces more NO than other isoforms in a wide range of cells and its induction usually occurs in an oxidative environment [14,16]. According to recent data, there are two main areas of nNOS localization in the brain. Firstly, nNOS is significantly expressed in the cytoplasm of some GABAergic neurons in the cortex and hippocampus [17,18]. Secondly, in situ hybridization showed the presence of neuronal NO synthase mRNA in the bodies of hippocampal pyramidal neurons [19], and experiments with electron microscopic immunohistochemistry demonstrated that many pyramidal excitatory neurons contain nNOS protein in their dendritic spines [20]. nNOS was also found in retinal cells [21] and spinal cord neurons [22]. Neuronal NO synthase is represented by several splice variants. The long isoform of neuronal NOS (αNOS1) contains a PDZ-binding domain that is capable of binding to the PDZ2 domain of synaptically located PSD95 protein [23,24]. Such binding contributes to the localization of nNOS in the postsynaptic density. There are also truncated enzyme variants known as βNOS1 and γNOS1 that lack the PDZ-binding domain. The βNOS1 isoform was found in some parts of the brain, in contrast to γNOS1, which is not found in the brain. [25]. Thus, 5–10% of αNOS1-expressing cells in the striatum and cortex also express the β-isoform, while some cells of the striatum only express βNOS1 [25].

NO plays key roles in maintaining hemostasis and the regulation of the activity of smooth muscle (especially the smooth muscles of blood vessels), neurons, and the gastrointestinal tract. It is involved in the regulation of all aspects of our life, including awakening, digestion, sexual function, perception of pain and pleasure, memory, and sleep. Its role in the formation of long-term potentiation has been shown many times both for the presynapse [17] and for the postsynapse [5,26]. NO is also important for learning [27,28,29] and memory consolidation and reconsolidation [30,31].

High concentrations of NO mediate neurotoxicity, leading to stroke and neurodegenerative diseases, such as Alzheimer’s and Parkinson’s. In cancer, low concentrations of NO contribute to tumor formation while high concentrations have a carcinogenic effect. There are several reviews of the involvement of NO in various pathological processes [10,16,32]. Most of the cytotoxicity associated with NO is due to peroxynitrite, produced from the reaction between NO and another free radical, the superoxide anion. Peroxynitrite interacts with lipids, DNA, and proteins via direct oxidative reactions or indirect mechanisms, and triggers cellular responses leading to cell necrosis or apoptosis [8]. The involvement of NO in neurodegenerative diseases is also important in the context of excitotoxicity, which refers to the loss of neurons caused by excessive activation of glutamate receptors. However, the effects of NO do not only depend on the concentration. Other factors, including the cell type and duration of exposure to NO, also play a significant role in determining the effect of NO [10].

NO production by nerve cells in response to excitatory stimuli was shown to require an influx of Ca^2+^, which implies that NOS activity is dependent on calcium and calmodulin [33]. The main postsynaptic stimulus for NO formation is activation of glutamate receptors and the corresponding Ca^2+^ influx. The produced NO can further modify glutamate channels, causing significant changes in their biochemical and electrophysiological properties. NO can modulate ionotropic glutamate receptors via direct or indirect pathways [34]. The direct route involves *S*-nitrosylation of AMPARs and NMDARs, and their associated proteins. *S*-nitrosylation is a physiologically important covalent binding of NO to the cysteine thiol group, which changes the protein’s physiological function. It was shown to play a large role in the signaling and regulation of glutamatergic synapses by modulating ion channel gating, membrane fusion, fission, and compartmentalization; or by protease-mediated protein degradation. For more information about *S*-nitrosylation, see the review [35]. The indirect pathway of NO effects starts with the stimulation of soluble guanylate cyclase (sGC), which leads to an increase in cGMP production and activation of cyclic guanosine monophosphate (cGMP)-dependent protein kinases [36]. NO can also indirectly control AMPAR subunit trafficking by different auxiliary proteins.

## 3. Nitric Oxide and AMPARs

The AMPA-type glutamate receptor is responsible for fast neurotransmission and aspects of activity-dependent synaptic plasticity thought to underlie higher-order cognitive functions, such as learning and memory [37,38,39].

AMPARs are expressed both in neurons and in glial cells of the central nervous system [40]. Most AMPARs are heterotetramers combined from the GluA1, GluA2, GluA3, and GluA4 subunits. Each subunit has a similar membrane topology and a basic structure containing about 900 amino acids, with a molecular weight of approximately 105 kDa. The terminal amino group is extracellular, and there are three membrane-spanning domains, one loop domain, and an intracellular domain with a carboxy-group at the end (C-terminal domain). This C-terminal domain is a highly variable part of the receptor that provides a platform for both protein interactions and post-translational modifications that regulate subunit-dependent transport [41,42].

Using immunogold staining and electron microscopy methods, Sans et al. showed that the number of GluA3 subunits in the structure of AMPARs is 10 times less than the number of GluA1 and GluA2 subunits [43]. Lu et al. demonstrated that ~80% of synaptic AMPARs in CA1 hippocampal neurons are GluA1-GluA2 heteromers [44]. However, other studies suggest that AMPARs are mainly heteromers of GluA1/GluA2 or GluA2/GluA3, with approximately equivalent amounts of each heteromeric complex in the hippocampus and rat cortex [45]. The GluA4 subunit, on the contrary, is less expressed in glutamatergic synapses of the adult brain [46].

In the adult brain, almost all GluA2 subunit mRNA undergoes post-transcriptional editing, which leads to a replacement of the neutral amino acid of glutamine with positively charged arginine in the polypeptide chain of the subunit. Such a replacement alters the electrophysiological properties of GluA2-containing AMPARs and makes them impermeable to calcium. [47]. GluA2-containing and GluA2-lacking AMPARs play different roles in synaptic plasticity, developmental mechanisms [48], and pathologies. As examples, the involvement of the GluA2-containing AMPARs was demonstrated in putative mechanisms of Alzheimer’s disease [49,50], blocking the interactions of GluA2 subunits with certain proteins inhibited mechanisms of neuropathic pain [51], and GluA2 subunit flip and flop isoforms are decreased in schizophrenia [52]. The involvement of GluA2-lacking AMPARs in pathologies is mainly due to their ability to conduct calcium ions, which is dangerous for cells at high concentrations. The number of these receptors increases in pilocarpine-induced status epilepticus [53,54], stroke and ischemia [55,56], cognitive impairment in diabetes [57], depression [58], and amyotrophic lateral sclerosis [59]. The involvement of calcium-impermeable (CI)- and calcium-permeable (CP)-AMPARs in synaptic plasticity differs functionally and temporally. LTP in CA1 hippocampal pyramidal neurons causes rapid incorporation of GluA2-lacking CP-AMPARs. CP-AMPARs are present transiently, being replaced by GluA2-containing AMPARs [60,61], and it is becoming increasingly obvious that the plastic changes of GluA2-lacking AMPARs also occur during various behavioral paradigms in vivo [62,63].

### 3.1. The Indirect cGMP-Dependent Pathway

One of the possible pathways of NO-mediated AMPARs subunit incorporation into the neuronal membrane is an indirect cGMP-dependent cascade (Figure 1). So far, it was found to be responsible for GluA1 subunit incorporation and can perhaps affect PKA-dependent GluA1 incorporation during synaptic plasticity. The AMPAR GluA1 subunit has a long cytosolic tail of ~80 amino acids, which provides a binding site for proteins and carries a class I PDZ binding motif at the C-terminus [64]. Cytosolic binding proteins include four PDZ domain proteins: SAP97 [65], mLin-10 [66], Shank3 [67], and nexin 27 [68]. In addition, this terminal domain binds, with non-PDZ bonds, to cytoplasmic proteins, including cGMP-dependent protein kinase II (cGKII or PKG), which regulates GluA1 subunit incorporation into the synaptic membrane, presumably in an NO-dependent manner. NO activates sGC, which initiates cGMP formation, and one of the targets of cGMP is cGKII [69]. Next, cGKII phosphorylates GluA1 at the S845 site, initiating its incorporation into the membrane [70,71]. In addition, stimulation of the CP-AMPARs in the striatum neurons caused activation of neuronal NO synthase and cGMP production that also led to S845 GluA1 subunit phosphorylation and insertion of the receptor into the membrane [72].

S845 of GluA1 is also a target for phosphorylation by cAMP-dependent protein kinase (PKA) and also controls GluA1 incorporation into the synaptic membrane [73]. Inhibiting PKA reduces the NOS activity and impairs the functioning of the glutamate-NO-cGMP pathway [74], which can normally lead to GluA1 subunit incorporation. Therefore, one can suggest that PKA and PKG act in parallel. Occasionally, they do act this way [75]; however, an increase in cGMP through the activity of nNOS constrains the induction of the PKA-dependent form of LTP induced by specific patterns of synaptic stimulation [76]. Moreover, another pattern of stimulation induces PKA-dependent LTP that also requires CP-AMPAR incorporation [77]. Considering the fact that both the cGMP-PKG pathway and PKA-dependent phosphorylation of S845 regulate the GluA1 subunit insertion, it is possible to speculate that NO activity specifically regulates the molecular basis of LTP induced by different types of stimulation protocols.

### 3.2. AMPARs S-Nitrosylation

NO also regulates the GluA1 subunit through a direct pathway, *S*-nitrosylation. A group of investigators showed that the cysteine-893 site of the GluA1 subunit can undergo *S*-nitrosylation and, using co-precipitation, the obtained data that suggest that the GluA1 subunit is linked to nNOS via SAP97 [78]. These data may explain the results of another study, where the authors found that overexpression of SAP97 in the synapses of cultured hippocampal neurons also increases the levels of PSD-95 and nNOS [79].

Another study demonstrated a signaling cascade leading to an increase in AMPAR conductivity and endocytosis: Calcium entry through NMDARs stimulates nNOS to produce NO that nitrosylates GluA1 at the C875 site. This event promotes phosphorylation of the S831 site, leading to increased channel conductivity. In addition, calcium influx via NMDARs regulates GluA1 endocytosis by binding the nitrosylated GluA1-C875 to the AP2 protein of the endocytotic machinery [80].

Thus, *S*-nitrosylation is an additional route of AMPAR subunit modulation (Figure 1), during which the GluA1 cysteines (C893 and C875) are modified by NO, affecting the channel conductance and subunit endocytosis. So far, this has only been demonstrated for the GluA1 subunit. *S*-nitrosylation can also regulate AMPAR trafficking by modifying AMPAR-associated proteins, which we will discuss below.

### 3.3. Protein–Protein Interactions

AMPARs’ incorporation into the synaptic membrane depends not only on the state of the receptor itself but also on interactions with various binding proteins. For instance, NO is involved in the regulation of GluA2 subunit incorporation into the membrane by nitrosylation of *N*-methylmaleimide-sensitive factor (NSF). NSF *S*-nitrosylation regulates exocytosis in cells [81] and, in particular, enhances NSF binding to GluA2, thereby increasing the surface expression of this subunit [82]. When NSF is unbound to GluA2 and GluA4 C-terminals, the amplitude of miniature EPSCs drops sharply, confirming the role of NSF in the functioning and distribution of AMPARs in the membrane [83]. Due to the ATPase activity of NSF [84], it likely causes declustering of the GluA2-PICK1 complex, resulting in the release of GluA2-containing AMPARs [84,85]. In addition, it has already been shown that PICK1 and NSF are necessary for embedding the GluA2 subunit, at least during plasticity associated with CP- to CI-AMPAR exchange [86]. One may assume that the same mechanism is also involved during LTP in GluA1−/− mice, since long-term potentiation in these mice is NOS dependent [87,88]; more precisely—αNOS1-dependent [89].

Regarding eNOS, it was suggested that eNOS could produce a tonic level of NO, which also affects plasticity [90]. In addition, phosphorylation of nNOS at the S1412 site by protein kinase B (Akt) increased the number of GluA2-containing AMPARs on the membrane surface, probably modulating the activity of this enzyme and, consequently, NO production [91]. Although the latter study did not analyze NSF nitrosylation, the results confirm NO involvement in the GluA2 distribution in the membrane. NSF-mediated GluA2 incorporation may be crucial for drug design since increased NSF–GluA2 interactions in the nucleus accumbens after withdrawal from cocaine attenuates the expression of behavioral sensitization and serves as a negative regulatory mechanism in drug-exposed individuals [92].

Nitrosylation can also affect proteins associated with the GluA1 subunit. These include, for example, proteins of the TARP family (transmembrane AMPAR regulatory proteins), a member of which is stargazin (TARP gamma 2) [93]. The interaction of AMPARs with stargazin regulates both the localization of the receptor on the synaptic membrane and the electrophysiological properties of the receptors, such as the conductivity and desensitization rate [94,95]. It was shown that NO can nitrosylate stargazin at cysteine-302. Nitrosylation of stargazin increases its binding to GluA1 and enhances the surface expression of the subunit [96]. Stimulation of NMDARs increased nitrosylation of stargazin and its interactions with GluA1 in an NO-dependent manner [96]. This mechanism may also be involved in AMPAR upregulation during cocaine sensitization since animals treated with injections of cocaine showed elevated stargazin and GluA1 surface expression [97].

Moreover, some studies revealed other nitrosylation target proteins that affect the number of certain AMPAR subunits in the membrane. *S*-nitrosylation of p35 protein leads to a decrease in the activity of the cyclin-dependent kinase 5 (Cdk5) [98], which entails the ubiquitination of PSD-95 protein [99]. According to other sources, this process leads to an increase in the number of GluA1 subunits in the synaptic membrane [100]. It is likely that the GluA1 internalization in this mechanism occurs due to PSD95 ubiquitination, which regulates both GluA1 and GluA2 surface expression [101]. Using various techniques, it was shown that NO regulates ubiquitin-dependent protein degradation [102,103]. Ubiquitin-dependent protein degradation, in turn, affects GluA1 and GluA2 AMPAR subunit trafficking [104], and, possibly, the p35-Cdk5 pathway is the link that completes the picture of intracellular ubiquitin-dependent cascades.

A large body of evidence demonstrates that nNOS activation can elicit the phosphorylation of cAMP response element-binding protein (CREB) [105,106,107]. During LTP, NO stimulates sGC and PKG, which act in parallel with PKA to increase CREB phosphorylation [75]. CREB is a transcription factor that binds to the cAMP-response element (CRE), a specialized stretch of DNA that is found within the regulatory region of numerous genes. Events at the neuronal membrane, which stimulate intracellular signaling cascades, cause the phosphorylation of CREB and activate it. Then, phosphorylation of CREB activates a cascade of events that involves the recruitment of associated proteins, such as CREB-binding protein (CBP), allowing it to switch certain genes on or off [108]. The ability of CREB to alter gene expression contributes to mechanisms linked to different patho- and physiological states, such as addiction, depression, memory facilitation, etc. For more information, see the review [109]. Thus, CREB phosphorylation was demonstrated to be necessary for synaptic maintenance and learning-induced changes of the AMPAR GluA1 subunit within the postsynaptic density (PSD), while it did not affect the levels of the GluA2/GluA3 subunits [110]. According to this fact, it is very likely that NO promotes the incorporation of at least the GluA1 subunit via CREB phosphorylation. CREB activation does not control GluA1 gene expression [110], although it possibly regulates the transcription of other proteins involved in AMPAR trafficking, such as brain-derived neurotrophic factor (BDNF) or the immediate early gene Arc [111,112]. Thereby, the mediators of such NO-dependent CREB activation remain unknown.

Additional components of the NO-dependent cascades are the extracellular-regulated kinases 1 and 2 (ERK1/2)—a subfamily of mitogen-activated protein kinases (MAPKs), which control embryogenesis, cell differentiation, cell proliferation, and cell death [113]. For instance, in rat cerebellar Purkinje cells, NO and PKG enhance phosphorylation of ERK1/2 and promote the declustering of GluA2/3 AMPARs [114]. NO can activate ERK1/2 either through activation of the monomeric G protein, Ras [115], or through activation of the sGC-cGMP-PKG pathway [116]. The latter NO–cGMP–PKG pathway has been shown to regulate CREB phosphorylation in the hippocampus partially by affecting ERK as an upstream kinase [75]; therefore, ERK1/2 seem to be an additional component of the NO-dependent cascade discussed above, leading to CREB activation and regulating GluA1 AMPAR surface expression. Regarding the first Ras-ERK1/2 pathway, a group of researchers suggested that it acts simultaneously with Ca^2+^/calmodulin kinase II (CaMKII) and the link between NOS and ERK1/2 appears only while learning and memory formation processes are occurring [117]. Previous studies have also suggested a link between the NO and ERK pathways while LTP formation occurs [118]. Moreover, it was reported that activated CaMKII stimulates Ras-ERK [119]. The Ras-ERK pathway, in turn, was shown to regulate activity-dependent exocytosis of AMPA receptors [120]. Thereby, ERK1/2 could be an element of another possible NO-dependent cascade, which leads to GluA1 subunit insertion into the membrane via CaMKII activity. Apparently, this cascade becomes active during memory formation. Although, the direct link between NO and ERK1/2-mediated GluA1 incorporation requires additional proofs.

Thus, besides the direct nitrosylation and the indirect cGMP-dependent pathway, there are several protein–protein interactions in which modulation by NO regulates AMPAR subunit trafficking. These include *S*-nitrosylation of NSF, stargazin, p35, and complicated pathway, which we summarized in Figure 2.

Interestingly, while NO production leads to changes in AMPAR subunit composition and surface expression through multiple biochemical cascades, AMPARs can themselves influence the NO synthesis. For instance, in hippocampal slices, using an NO sensor, it was shown that even with blocked NMDARs, the level of calcium ions in the cell is sufficient to trigger nNOS activity, since the amount of free NO was greater than in the case with simultaneously blocked NMDA and AMPARs. Moreover, the blockade of calcium-permeable AMPARs with PhTx-433 reduced the NO production [121], which directly indicates the dependence of NOS on free calcium ions entering not only through the NMDA channels but also through the calcium-permeable AMPARs. This AMPAR-induced calcium influx probably occurs through L-type voltage-gated calcium channels (VDCC) [122]. The latter was demonstrated by using transgenic mice ubiquitously expressing a Förster resonance energy transfer (FRET)-based cGMP indicator to analyze the cGMP responses induced by NO or glutamatergic agonists in cultured hippocampal and cortical neurons. In this study, besides the well-established NMDA-induced cGMP responses, the glutamatergic agonist AMPA independently elicited cGMP signals in hippocampal and cortical neurons by triggering L-type VGCC-dependent Ca^2+^ influx [122]. This finding is consistent with previous studies demonstrating that nNOS can be activated by Ca^2+^ influx through VDCCs [123] and CP-AMPARs [124]. This postsynaptic calcium influx can also mediate the activation of nNOS in the postsynapse and define presynaptic expression of the spike-timing-dependent LTP (tLTP) in glutamatergic synapses of the dorsal raphe nucleus via the cGMP/PKG pathway. This tLTP is independent of NMDA receptors but requires activation of the CP-AMPA receptors and VDCCs [125]. It was also shown that AMPAR can influence the NO-dependent cascades in the retina cells in a specific way, which we will discuss in the next paragraph.

### 3.4. NO and AMPARs in Retina as an Example of Mutual Interaction

In the visual system, NO is involved in the processing of visual information from the retina to the higher visual centers. There is a comprehensive review by Lima et al. discussing the role of NO in the visual system [126]. In the retinal cells, it was shown that not only NO influences the expression of AMPAR subunits, but also it was shown that AMPARs can influence the NO production. We focus on the retina and review the mechanisms of cooperation of NO and AMPA receptors in retinal cells, because this cooperation reflects the multifunctionality of NO well even in such a strictly specialized group of cells. In the retina, NOS is present in different types of cells, including the pigment epithelium [127], photoreceptors, Müller cells, horizontal, bipolar, amacrine, and ganglion cells [128]. In these cells, NO initially modulates the signal generally by altering the cGMP levels [126]. Thus, some studies demonstrated the involvement of NO through activation of the sGC-cGMP-PKG pathway, acting as a modulator of the temporal properties of the glutamate response [129,130]. Perfusion of acute slices of rat retina with the NO precursor L-arginine shortened the AMPA receptor-dependent component of the glutamate response and did not change the kainate receptor-dependent component in bipolar cells [129]. In addition, the NO/cGMP pathway modulates AMPARs in the cultured horizontal cell, reducing their responsiveness to agonists [131]. Later, experiments with the concentration ramp technique showed that this effect is due to a decrease of AMPARs’ affinity for glutamate by NO. In these experiments, NO simultaneously increased AMPARs’ maximal current at high glutamate concentrations, suggesting that NO is a candidate for mediating adaptational changes in retinal neurons and synapses [132]. The exact mechanism of such modulation remains unknown.

In retinal neurons, NO-AMPAR GluA1 coupling may simultaneously influence the cell apoptosis and viability: Signaling mediated by the CP-AMPAR activation increases the nNOS activity, and modulates NO production, leading to cellular death [124]. However, NO increases CREB activity via the PKG pathway [133], and prolonged phosphorylation of CREB prevents cell apoptosis induced by CP-AMPAR-mediated excitotoxicity [134].

In addition to this protective effect of CREB through NO generation, the same CP-AMPAR-dependent nNOS activation is able to cause cell death through upregulation of another important protein—the nonreceptor tyrosine kinase Src. Src has two specific phosphorylation sites that regulate its activity within a cell—Tyr416 and Tyr527. When Tyr416 is phosphorylated by Shp-1 (Src homology region 2 domain-containing tyrosine phosphatase 1) or by Shp-2, Src is rendered in its active conformation, resulting in full catalytic activation [135]. Elevated NO levels through a cGMP- and PKG-dependent manner induce Src phosphorylation at the Tyr416 site and dephosphorylation at Tyr527, leading to Src activation and apoptosis in cultured retinal neurons [124]. In this mechanism, ERK acts as a downstream target of AMPA-induced Src activation and NO signaling [124,136]. Interestingly, in one study, the ERK activation was shown to cause the CREB phosphorylation described above [133], which prevented CP-AMPARs-induced excitotoxicity [134], while in another ERK phosphorylation through the same NO-PKG cascade, the mechanism of retinal cell apoptosis is involved [124]. Apparently, there are additional proteins in this cascade determining the fate of the cell.

Summarizing, in retinal cells, NO can simultaneously serve as a modulator of AMPARs signal and participate in the molecular cascades leading to cell death or to its survival during CP-AMPARs-induced excitotoxicity. Further study of these interactions can contribute to understanding the molecular mechanisms of vision.

## 4. Polyamines and NO

On the inner side of the membrane, CP-AMPARs can be blocked by polyamines, such as spermine and spermidine. At positive membrane potentials, the polyamine block prevents outgoing currents, forming a rectified current–voltage curve for these receptors [137,138]. Depending on the membrane potential, opening the channel leads either to release from the block, or to blockade of the ion flow [139]. Polyamines are organic compounds with more than two amino groups (NH_3_^+^) and are present in eukaryotic and prokaryotic cells [140,141]. Polyamines are synthesized in almost all mammalian cells from amino acids, such as arginine, ornithine, and methionine [142]. Arginine can also be metabolized to NO via NOS. Intracellular arginine levels limit the rate of production of ornithine by arginase, and the rate of production of NO by NOS [143], resulting in competition for intracellular arginine. In cells that express both nNOS and arginase 1, competition for arginine reduces NO production due to overexpression of arginase [144,145,146]. In addition, in one of the studies in cerebellar cells, the authors showed that endogenous polyamines do not affect the activity of NO synthase. This is in contrast to exogenous polyamines, which inhibit the conversion of [3H] L-arginine to [3H] L-citrulline by NO synthase. The ability of polyamines to interact with NO synthase is strongly influenced by the concentration of L-arginine and NALTPH [147]. This might be crucial during whole-cell patch-clamp experiments when the concentration of polyamines in the patch pipette can influence the NO synthase activity and may lead to distorted results. Given this downregulation of NO production by polyamines, the inhibition of NO synthase, in turn, might affect polyamine generation. For example, during inflammation, there is a switch from high NO to low NO formation, and an increase of L-ornithine and polyamine synthesis during the tissue repair and cell proliferation [148]. Thus, since polyamines determine the conductance of CP-AMPARs, NO may be involved in determining the physiological properties of these receptors by regulating polyamine production in the cells.

In general, NO affects the incorporation of various AMPAR subunits in a variety of ways, from direct nitrosylation of the receptor subunits mediated by the NO-PKG-cGMP protein kinase G receptor phosphorylation cascade, to various modifications of AMPAR-binding proteins. This modulation can also affect AMPAR functions, such as the permeability, by NO-dependent modifications, and many others that determine the unique profile of the functioning of AMPARs in different synapses and at different times, regulating synaptic transmission in the brain [149].

## 5. NO and Memory Labilization/Erasure during Reconsolidation

Many studies suggested that NO is involved in the molecular mechanisms of memory formation [29,150,151]. The concept of memory reconsolidation assumes that newly acquired memories are not simply consolidated once and then remain intact forever but can be labilized by retrieval and then consolidated again [152]. Many studies demonstrated in animals from mollusks to higher vertebrates that after presentation of a specific reminder, the reactivated “old” memories become labile and again susceptible to such amnesic agents as protein synthesis blockers, etc. This re-stabilization phase is usually referred to as reconsolidation [153,154]. One of the hypotheses suggests that there are local effects of the NO generated by Ca^2+^ influx via activated NMDA receptors in neurons of the network activated by a specific reminder [155]. Investigation of the NO function in the reconsolidation of long-term context memory in terrestrial snails (*Helix lucorum* L.) showed that if the retrieval was performed with the protein synthesis blocker anisomycin present, the context memory was impaired when tested 24 h and later, whereas the retrieval under combined injection of anisomycin and NO-synthase inhibitors, or an NO scavenger showed no impairment of long-term context memory. These results suggest that NO is necessary for labilization/erasure of a consolidated context memory [30].

It was shown that injection of the selective blockers of NO-synthase (3-Br-7-Ni or ARL) in behavioral experiments in rats under conditions of memory reactivation and blockade of protein synthesis rescues destabilization of the conditioned fear memory [31]. These results demonstrate that disruption of the NO signaling pathway during memory reconsolidation can prevent changes in long-term memory and confirm the role of nitric oxide in the destabilization of existing fear memory retrieved by reminding [31].

AMPA receptors, as major elements of excitatory transmission in the mammalian central nervous system, are also strongly involved in memory modifications and maintenance. AMPA receptors’ trafficking is involved in memory expression, and regulation of the trafficking is very important for memory storage and modifications [156,157,158]. Blockade of AMPA receptors’ endocytosis in the postsynapse leads to the preservation of memory [62].

Other evidence comes from electrophysiological experiments with long-term potentiation. It was shown that simultaneous blockade of NO synthesis and protein synthesis prevents the LTP decline normally induced by NO blockade [102] or a protein synthesis blockade [26]. It should be noted that different protocols of stimulation can induce NO-dependent and “NO-independent” forms of LTP. Interestingly, simultaneous blockade of sGC and protein synthesis did not prevent a decrease in LTP [26]. It suggests that the effect of “LTP rescue” is not mediated by the NO-sGC-PKG pathway and is possibly mediated by nitrosylation.

AMPA receptors’ trafficking underlies LTP maintenance, and the mechanisms of trafficking regulation are proposed to be different in the early and late phases of LTP [149,159,160]. Based on these data, we suggest that NO is involved in the regulation of AMPA receptors’ trafficking during LTP and memory modifications.

## 6. Summarizing and Concluding Remarks

In the present report, we reviewed the general pathways by which NO interacts with AMPA receptors, influences their properties, and regulates their surface expression. NO can modulate these receptors via direct (*S*-nitrosylation) or indirect (cGMP-dependent) pathways. AMPA receptors have cysteines that can undergo functionally significant nitrosylation. In addition, modifications may impact not only the receptors themselves but also their auxiliary proteins.

Thus, NO can act via direct nitrosylation of the GluA1 subunit, and for some brain regions, it was shown that the GluA1 subunit is colocalized with neuronal NO synthase [161]. Besides direct nitrosylation, NO also comes into cooperation with other intracellular molecules, thereby controlling AMPARs subunits “from a distance”. The classical NO-cGMP pathway and different auxiliary proteins, such as stargazin, CREB, and ERK, can play the role of such mediator molecules. Nitrosylation of NSF promotes surface expression of the GluA2 subunits. In addition, NO production presumably determines the permeability of CP-AMPARs due to their sensitivity to polyamines. Moreover, the competition for intracellular arginine in the production of polyamines and NO is an important mechanism that should be taken into account when performing whole-cell recordings. NO–AMPAR connections have been demonstrated in different brain areas from the hippocampus to the visual cortex. In retinal cells, NO can simultaneously serve as a modulator of AMPARs’ signal and participate in the molecular cascades leading to cell death or to its survival during CP-AMPARs-induced excitotoxicity.

Since both NO and AMPARs are involved in multiple aspects of cell life, it is hardly surprising that the relationships between these two molecules have been observed during different physiological and pathophysiological states, such as plasticity [122,125], excitotoxicity [124], hypoxia [162], the effect of cocaine [92,97], inflammation [148], and so on. To conclude, in this review, we demonstrated that two important factors involved in a broad range of cellular mechanisms—NO and AMPARs—are closely interconnected in various aspects of neuronal functioning.

## Figures and Tables

**Figure 1 ijms-21-00981-f001:**
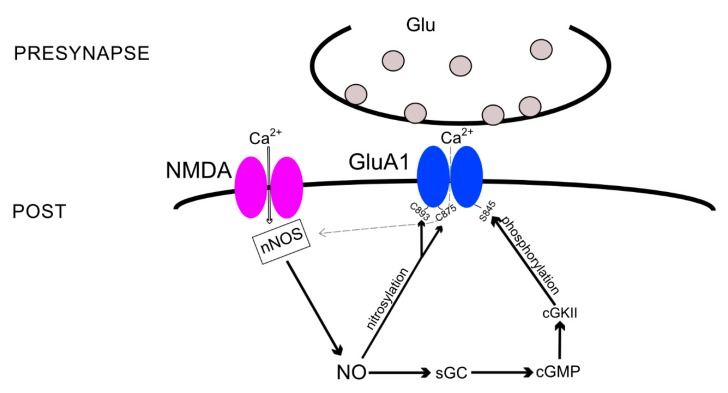
Schematic presentation of indirect cyclic guanosine monophosphate (cGMP)-dependent pathway and direct nitrosylation of glutamate ionotropic receptor AMPA type subunit 1 (GluA1) regulating its incorporation into the synaptic membrane. Calcium influx through *N*-methyl-d-aspartate (NMDA) or calcium-permeable AMPA (CP-AMPA) receptors activates the neuronal NO-synthase (nNOS) to produce nitric oxide (NO), which can either directly nitrosylate GluA1’s C893 [78] or C875 [80], causing the subunit’s insertion into the membrane or phosphorylates the GluA1’s S845 via the soluble guanylate cyclase (sGC)-cGMP-cGMP-dependent protein kinase II (cGKII) pathway activation [69,70,71].

**Figure 2 ijms-21-00981-f002:**
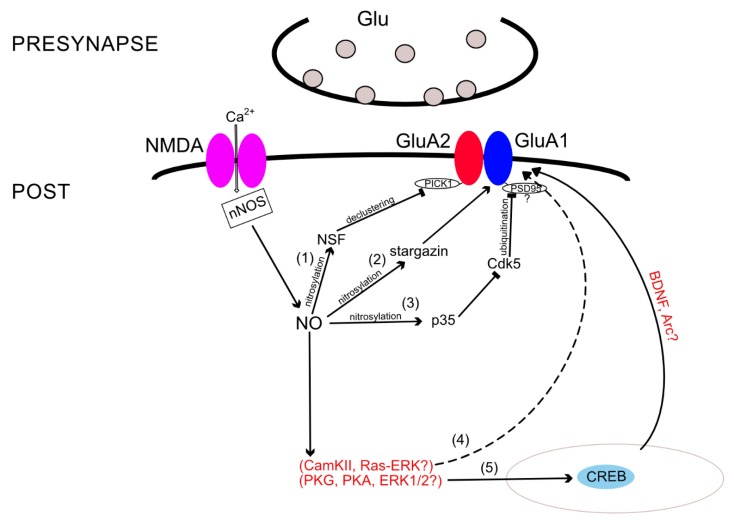
Proposed model of NO-dependent protein–protein interactions underlying GluA1 and GluA2 subunits’ trafficking. Red text and black dashed line represent cascades requiring additional proofs. (1) *S*-nitrosylation of *N*-methylmaleimide-sensitive factor (NSF) causes declustering of the protein interacting with C kinase 1) (PICK1)-GluA2 complex, which promotes incorporation in the membrane of GluA2-containing AMPA receptors [84,85]; (2) nitrosylation of stargazin increases its binding to GluA1 and enhances the surface expression of the subunit [96]; (3) NO nitrosylates the p35 protein, leading to a decrease in the activity of the cyclin-dependent kinase 5 (Cdk5) enzyme [98]. This process leads to an increase in the number of GluA1 subunits in the synaptic membrane [100], probably through the ubiquitination of postsynaptic density protein 95 (PSD-95) protein [99]; (4) a potential NO-dependent cascade resulting in the GluA1 insertion. The link between NOS and the Ras-extracellular signal-regulated kinase (ERK) pathway can appear during long-term potentiation (LTP), possibly via Ca^2+^/calmodulin-dependent protein kinase II (CaMKII) activation [117,118,119,120]; (5) an additional prospective cAMP response element-binding protein (CREB)-mediated pathway of GluA1 incorporation. NO, through a number of possible proteins, promotes phosphorylation of CREB [105,106,107], which, in turn, increases the GluA1 membrane expression [110].

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
