# Peer review of "Modulation of AMPA Receptors by Nitric Oxide in Nerve Cells"

_ijms, 2020, doi:10.3390/ijms21030981_

Round 1

Reviewer 1 Report

The authors address an interesting and important topic in the review, namely the impact of NO on the function of ligand gated glutamate receptors. The authors introduced the GluR subtypes and the effects of NO on these receptors. Generally the presented information are all correct, as far as I can judge. However, the manuscript suffers from an ever present superficial description of scientific facts, as I mentioned e.g. in line 265.
On the other hand some parts are overloaded with details. Take together, the manuscript, was at least for me, hard to read. Overall I am missing a concise presentation of the reviewed data. Maybe the manuscript can gain some precision if the others focus only on AMPA receptors and the role of AMPAR in NOsignallng. I am aware of the fact that the function/action of AMPAR and NMDAR can be tightly connected.However I would appreciate a more concise presentation of data link to the AMPAR, as announced by the title of the review.

Minor points.
The manuscript should undergo a thorough language check and format check.

Author Response

Reviewer 1

«The authors address an interesting and important topic in the review, namely the impact of NO on the function of ligand gated glutamate receptors. The authors introduced the GluR subtypes and the effects of NO on these receptors. Generally the presented information are all correct, as far as I can judge. However, the manuscript suffers from an ever present superficial description of scientific facts, as I mentioned e.g. in line 265.
On the other hand some parts are overloaded with details. Take together, the manuscript, was at least for me, hard to read. Overall I am missing a concise presentation of the reviewed data. Maybe the manuscript can gain some precision if the others focus only on AMPA receptors and the role of AMPAR in NO signallng. I am aware of the fact that the function/action of AMPAR and NMDAR can be tightly connected. However I would appreciate a more concise presentation of data link to the AMPAR, as announced by the title of the review.
Minor points.
The manuscript should undergo a thorough language check and format check.»

Response: 

Dear Reviewer,

We have significantly reworded the text, made it more concise according to your recommendations, and it was checked by a native English speaker (neurobiologist D. Jappy), a part concerning NMDA receptors was removed.

With gratitude for detailed comments on the text,

Ivanova, P. Balaban, N. Bal.

Reviewer 2 Report

This review by Violetta Ivanova and colleagues is interesting and well-done in my opinion. 

Just a couple of modifications would contribute to improve the readability of the paper: 

as the involvement of NO and glutamate receptors is relevant for glutamate excitotoxicity and neurodegenerative diseases, and appropriately discussed in the manuscript, I suggest to cite this important question also in the abstract; I suggest to include one or two figures summarizing the mechanisms of NO and glutamate receptor interactions; the english is good but a mild revision and grammar check would be helpful.

Author Response

Dear Reviewer,

please find our responses below.

Point 1. As the involvement of NO and glutamate receptors is relevant for glutamate excitotoxicity and neurodegenerative diseases, and appropriately discussed in the manuscript, I suggest to cite this important question also in the abstract.

Response 1: We included the following sentence to the abstract as suggested. "Interactions of NO and AMPA receptors were observed in important phenomena such as glutamatergic excitotoxicity in retinal cells, synaptic plasticity, and neuropathologies"

Point 2. I suggest to include one or two figures summarizing the mechanisms of NO and glutamate receptor interactions.

Response 2: We included 2 figures to the manuscript according to your comment.

Point 3.  The english is good but a mild revision and grammar check would be helpful.

Response 3: The manuscript was checked by a native English speaker (neurobiologist D. Jappy).

With gratitude for detailed comments on the text,

Ivanova, P. Balaban, N. Bal.

Round 2

Reviewer 1 Report

Due to the fact that the authors focus now on AMPA-R and the impact of NO on these receptors, the manuscript is more concise. The figures add value to the manuscript, however I am suggesting to be more precise with the caption and the figure legends should be edited. The legends should at least contain a short explanation of what is shown.
The paragraph about the retina appears somehow „lost“. I did not understand the rational to add the given information in a separate paragraph, especially as the paragraph does not contain a significant amount new information.

Author Response

Dear Reviewer!

In accordance with your comments, we enlarged legends, re-structured the text concerning retina (eliminated it as a separate item), only necessary information is present now in «NO and AMPAR». Retina cells demonstrate unique interactions of NO and AMPARs that stress the multifunctional nature of NO, and we feel it necessary to mention in this review.

With respect and gratitude,

V.Ivanova, P.Balaban, N.Bal.